# Generalizing Few-Shot Named Entity Recognizers to Unseen Domains with Type-Related Features

**Zihan Wang**[1,2], **Ziqi Zhao**[1], **Zhumin Chen**[1], **Pengjie Ren**[1],
**Maarten de Rijke**[2] and **Zhaochun Ren**[3*]

[1]Shandong University, Qingdao, China
[2]University of Amsterdam, Amsterdam, The Netherlands
[3]Leiden University, Leiden, The Netherlands
{zihanwang.sdu,ziqizhao.work}@gmail.com, chenzhumin@sdu.edu.cn
jay.ren@outlook.com, m.derijke@uva.nl, z.ren@liacs.leidenuniv.nl

## Abstract

Few-shot named entity recognition (NER) has shown remarkable progress in identifying entities in low-resource domains. However, few-shot NER methods still struggle with out-of-domain (OOD) examples due to their reliance on manual labeling for the target domain. To address this limitation, recent studies enable generalization to an unseen target domain with only a few labeled examples using data augmentation techniques. Two important challenges remain: First, augmentation is limited to the training data, resulting in minimal overlap between the generated data and OOD examples. Second, knowledge transfer is implicit and insufficient, severely hindering model generalizability and the integration of knowledge from the source domain. In this paper, we propose a framework, *prompt learning with type-related features* (PLTR), to address these challenges. To identify useful knowledge in the source domain and enhance knowledge transfer, PLTR automatically extracts entity type-related features (TRFs) based on mutual information criteria. To bridge the gap between training and OOD data, PLTR generates a unique prompt for each unseen example by selecting relevant TRFs. We show that PLTR achieves significant performance improvements on in-domain and cross-domain datasets. The use of PLTR facilitates model adaptation and increases representation similarities between the source and unseen domains.[1]

## 1 Introduction

Named entity recognition (NER) aims to detect named entities in natural languages, such as locations, organizations, and persons, in input text (Zhang et al., 2022; Sang and Meulder, 2003; Yang et al., 2017). This task has gained significant attention from both academia and industry due to its wide range of uses, such as question answering and document parsing, serving as a crucial component in natural language understanding (Nadeau and Sekine, 2007; Ma and Hovy, 2016; Cui and Zhang, 2019; Yamada et al., 2020). The availability of labeled data for NER is limited to specific domains, leading to challenges for generalizing models to new domains (Lee et al., 2022; Cui et al., 2021; Ma et al., 2022).

To overcome this issue, recent research focuses on enabling models to effectively learn from a few labeled examples in new target domains (Lee et al., 2022; Ma et al., 2022; Das et al., 2022; Chen et al., 2022a; Wang et al., 2022, 2023) or on exploring data augmentation techniques, leveraging automatically generated labeled examples to enrich the training data (Zeng et al., 2020). However, these methods still require manual labeling for target domains, limiting their applicability in zero-shot scenarios with diverse domains.

Recently, Yang et al. (2022) have explored a new task, *few-shot cross-domain* NER, aiming to generalize an entity recognizer to unseen target domains using a small number of labeled in-domain examples. To accomplish this task, a data augmentation technique, named FactMix, has been devised. FactMix generates semi-fact examples by replacing the original entity or non-entity words in training instances, capturing the dependencies between entities and their surrounding context. Despite its success, FactMix faces two challenges:

**Augmentation is limited to the training data.** Since the target domain is not accessible during training, FactMix exclusively augments the training data from the source domain. As a result, there is minimal overlap between the generated examples and the test instances at both the entity and context levels. For instance, only 11.11% of the entity words appear simultaneously in both the generated data (by FactMix) and the AI dataset (target domain). At the context level, as demonstrated

---
*Corresponding author.
[1]Our code is available at https://github.com/WZH-NLP/PLTR.

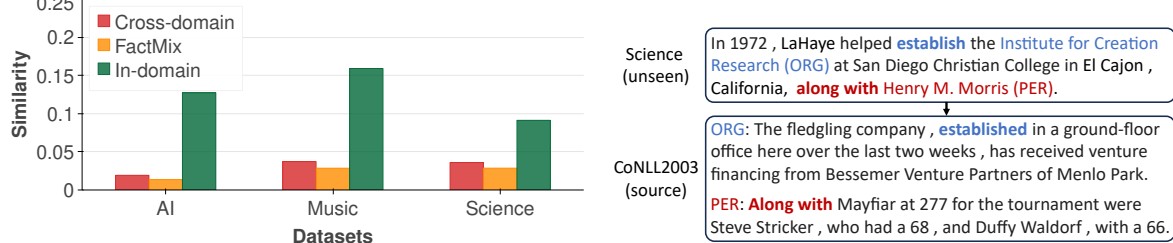

(a) Average similarities between pairs of sentences.

(b) Type-related features in the source domain.

Figure 1: (a) Average SBERT similarities (Reimers and Gurevych, 2019) between pairs of sentences that contain the same type of entities. The source domain dataset is CoNLL2003 (Sang and Meulder, 2003); the target domain datasets include AI, Music, and Science (Liu et al., 2021). In the "Cross-domain" setting, one sentence is from the source domain and the other is from the target domain. In the "FactMix" setting, one sentence is from augmented data by FactMix (Yang et al., 2022), and the other is from the target domain. In the "In-domain" setting, both sentences are from the target domain. (b) Examples of type-related features in the source domain.

in Fig. 1(a), the average sentence similarity between the augmented instances and the test examples is remarkably low. These gaps pose severe challenges in extrapolating the model to OOD data. To address this problem, we incorporate natural language prompts to guide the model during both training and inference processes, mitigating the gap between the source and unseen domains.

**Knowledge transfer is implicit and insufficient.** Intuitively, better generalization to unseen domains can be accomplished by incorporating knowledge from the source domain (Ben-David et al., 2022). However, in FactMix, the transfer of knowledge from the source domain occurs implicitly at the representation level of pre-trained language models. FactMix is unable to explicitly identify the type-related features (TRFs), i.e., tokens strongly associated with entity types, which play a crucial role in generalization. E.g., as illustrated in Fig. 1(b), the words "established" and "along with" exhibit a close relationship with organization and person entities, respectively, in both domains. This knowledge can greatly assist in recognizing organizations and persons in the target domain.

To tackle this limitation, we introduce mutual information criteria to extract informative TRFs from the source domain. Furthermore, we construct a unique prompt for each unseen instance by selecting relevant TRFs. Intuitively, these generated prompts serve as distinctive signatures, linking unfamiliar examples to the knowledge within the source domain.

**Contributions.** In this paper, we present a framework, named *prompt learning with type-related features* (PLTR) for few-shot cross-domain NER,

to effectively leverage knowledge from the source domain and bridge the gap between training and unseen data. As Fig. 2 shows, PLTR is composed of two main phases: (i) type-related feature extraction, and (ii) prompt generation and incorporation. To identify valuable knowledge in the source domain, PLTR uses mutual information criteria to extract entity type-related features (TRFs). PLTR implements a two-stage framework to mitigate the gap between training and OOD data. Firstly, given a new example, PLTR constructs a unique sequence by selecting relevant TRFs from the source domain. Then, the constructed sequences serve as prompts for performing entity recognition on the unseen data. Finally, a multi-task training strategy is employed to enable parameter sharing between the prompt generation and entity recognition. Similar to FactMix, PLTR is a fully automatic method that does not rely on external data or human interventions. PLTR is able to seamlessly integrate with different few-shot NER methods, including standard fine-tuning and prompt-tuning approaches.

In summary, our contributions are: (i) to the best of our knowledge, ours is the first work to study prompt learning for few-shot cross-domain NER; (ii) we develop a mutual information-based approach to identify important entity type-related features from the source domain; (iii) we design a two-stage scheme that generates and incorporates a prompt that is highly relevant to the source domain for each new example, effectively mitigating the gap between source and unseen domains; and (iv) experimental results show that our proposed PLTR achieves state-of-the-art performance on both in-domain and cross-domain datasets.

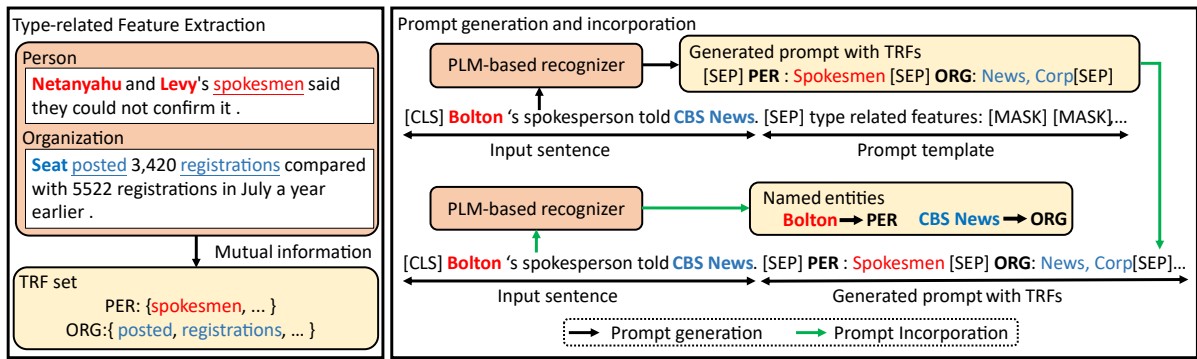

Figure 2: An overview of PLTR. PLTR has two main phases: (i) type-related feature extraction, and (ii) prompt generation and incorporation. Besides, we utilize a multi-task training strategy to enable parameter sharing between prompt generation and incorporation.

## 2 Related work

**Cross-domain NER.** The task of cross-domain NER aims to transfer NER models across diverse text styles (Pan et al., 2013; Liu et al., 2021; Chen et al., 2021; Lee et al., 2018; Yang et al., 2017; Jia et al., 2019; Jia and Zhang, 2020; Zheng et al., 2022; Zhang et al., 2022; Hu et al., 2023; Wang et al., 2021). Yang et al. (2017) train NER models jointly in the source and target domains, while Jia et al. (2019) and Jia and Zhang (2020) leverage language models for cross-domain knowledge transfer. Zhang et al. (2022) introduce a modular learning approach that decomposes NER into entity span detection and type classification subtasks. However, these methods still rely on NER annotations or raw data in the target domain.

**Few-shot NER and prompt-based learning.** Few-shot named entity recognition (NER) is the task of identifying predefined named entities using only a small number of labeled examples (Wiseman and Stratos, 2019; Yang and Katiyar, 2020; Das et al., 2022; Zeng et al., 2020; Ma et al., 2023). Various approaches have been proposed to address this task. For instance, Huang et al. (2021) investigate the effectiveness of self-training methods on external data using distance-based approaches, where the label of the nearest neighbors is copied. Zeng et al. (2020) involves generating counterfactual examples through interventions to augment the original dataset. Additionally, prompt-based learning, which has gained prominence in natural language processing, has also been applied to few-shot NER (Cui et al., 2021; Ma et al., 2022; Lee et al., 2022; Das et al., 2022; Chen et al., 2022b; Li et al., 2022; Dong et al., 2023; Fang et al., 2023). In particular, Das et al. (2022) incorporate contrastive

learning techniques with prompts to better capture label dependencies. Furthermore, Ma et al. (2022) develop a template-free approach to prompt NER, employing an entity-oriented objective. Recently, several studies have conducted analyses of the performance of current large language models (LLMs), such as the GPT series (Brown et al., 2020; OpenAI, 2023), in the context of the few-shot NER task (Gutierrez et al., 2022; Han et al., 2023; Sun et al., 2023). Nevertheless, these investigations have revealed a substantial performance gap between recent LLMs and state-of-the-art methods. Consequently, due to their high running costs and underwhelming performance, we do not consider recent LLMs as the basic model of our proposed framework (refer to Sec. 3.2). As mentioned in Sec. 1, previous few-shot NER methods primarily focus on in-domain settings and require manual annotations for the target domain, which poses a challenge for generalizing to OOD examples.

The field of **few-shot cross-domain learning** is inspired by the rapid learning capability of humans to recognize object categories with limited examples, known as rationale-based learning (Brown et al., 2020; Shen et al., 2021; Chen et al., 2022a; Baxter, 2000; Zhang et al., 2020). In the context of NER, Yang et al. (2022) introduce the few-shot cross-domain setting and propose a two-step rationale-centric data augmentation method, named FactMix, to enhance the model's generalization ability.

In this paper, we focus on few-shot cross-domain NER. The most closely related work is Fact-Mix (Yang et al., 2022). FactMix faces two challenging problems: (i) augmentation is limited to the training data, and (ii) the transfer of knowledge from the source domain is implicit and insufficient.

In our proposed PLTR, to identify useful knowledge in the source domain, mutual information criteria are designed for automatic type-related feature (TRF) extraction. In addition, PLTR generates a unique prompt for each unseen example based on relevant TRFs, aiming to reduce the gap between the source and unseen domains.

## 3 Preliminaries

### 3.1 Task settings

A NER system takes a sentence $\mathbf{x} = x_1, \ldots, x_n$ as input, where $\mathbf{x}$ is a sequence of $n$ words. It produces a sequence of NER labels $\mathbf{y} = y_1, \ldots, y_n$, where each $y_i$ belongs to the label set $\mathcal{Y}$ selected from predefined tags $\{B_t, I_t, S_t, E_t, O\}$. The labels $B$, $I$, $E$, and $S$ indicate the beginning, middle, ending, and single-word entities, respectively. The entity type is denoted by $t \in \mathcal{T} = \{\text{PER}, \text{LOC}, \text{ORG}, \text{MISC}, \ldots\}$, while $O$ denotes non-entity tokens. The source dataset and out-of-domain dataset are represented by $\mathcal{D}_{in}$ and $\mathcal{D}_{ood}$, respectively. Following Yang et al. (2022), we consider two settings in our task, the *in-domain setting* and the *out-of-domain (OOD) setting*. Specifically, we first train a model $\mathcal{M}_{in}$ using a small set of labeled instances from $\mathcal{D}_{in}$. Then, for in-domain and OOD settings, we evaluate the performance of $\mathcal{M}_{in}$ on $\mathcal{D}_{in}$ and $\mathcal{D}_{ood}$, respectively.

### 3.2 Basic models

Since our proposed PLTR is designed to be model-agnostic, we choose two popular NER methods, namely standard fine-tuning and prompt-tuning respectively, as our basic models. As mentioned in Sec. 2, due to their high costs and inferior performance on the NER task, we do not consider recent large language models (e.g., GPT series) as our basic models.

**Standard fine-tuning method.** We employ pre-trained language models (PLMs) such as BERT (Devlin et al., 2019) and RoBERTa (Liu et al., 2019) to generate contextualized word embeddings. These embeddings are then input into a linear classifier with a softmax function to predict the probability distribution of entity types. The process involves feeding the input token $x$ into the feature encoder PLM to obtain the corresponding contextualized word embeddings $\mathbf{h}$:

$$\mathbf{h} = \text{PLM}(x), \qquad (1)$$

where $\mathbf{h}$ represents the sequence of contextualized

word embeddings derived from the pre-trained language models. To recognize entities, we optimize the cross-entropy loss $\mathcal{L}_{NER}$ as:

$$\mathcal{L}_{NER} = -\sum_{c=1}^{N} y_{o,c} \log (p_{o,c}), \qquad (2)$$

where $N$ denotes the number of classes, $y$ is a binary indicator (0 or 1) indicating whether the gold label $c$ is the correct prediction for observation $o$, and $p$ is the predicted probability of $c$ for $o$.

**Prompt-tuning method.** The prompt-tuning method for NER tasks involves the use of mask-and-infill techniques based on human-defined templates to generate label words. We adopt the recent EntLM model proposed by Ma et al. (2022) as our benchmark for this method. First, a label word set $\mathcal{V}_l$ is constructed through label word engineering, which is connected to the label set using a mapping function $\mathcal{M} : \mathcal{Y} \rightarrow \mathcal{V}_l$. Next, entity tokens at entity positions are replaced with the corresponding label word $\mathcal{M}(y_i)$. The resulting modified input is then denoted as $\mathbf{x^{Ent}} = \{x_1, \ldots, \mathcal{M}(y_i), \ldots, x_n\}$. The language model is trained by maximizing the probability $P(\mathbf{x}^{Ent} \mid \mathbf{x})$. The loss function for generating the prompt and performing NER is formulated as:

$$\mathcal{L}_{NER} = -\sum_{i=1}^{N} \log P(x_i = x_i^{Ent} \mid \mathbf{x}), \quad (3)$$

where $N$ represents the number of classes. The initial parameters of the predictive model are obtained from PLMs.

## 4 Method

In this section, we present the two primary phases of the proposed PLTR method, as depicted in Fig. 2: (i) type-related feature extraction (see Sec. 4.1), and (ii) prompt generation and incorporation (see Sec. 4.2).

### 4.1 Type-related feature extraction

As mentioned in Sec. 1, type-related features (TRFs), which are tokens strongly associated with entity types, play a crucial role in the few-shot cross-domain NER task. To extract these features, we propose a mutual information based method for identifying TRFs from the source domain. Here, we define $\mathcal{S}_i$ as a set that contains all sentences from the source domain where entities of the $i$-th type appear, and $\mathcal{S} \backslash \mathcal{S}_i$ as a set that contains

sentences without entities of the $i$-th type. In our method, we consider a binary variable that indicates examples (texts) from $\mathcal{S}_i$ as 1, and examples from $\mathcal{S}\backslash\mathcal{S}_i$ as 0. To find tokens closely related to $\mathcal{S}_i$, we first calculate the mutual information between all tokens and this binary variable, and then select the top $l$ tokens with the highest mutual information scores. However, the mutual information criteria may favor tokens that are highly associated with $\mathcal{S}\backslash\mathcal{S}_i$ rather than with $\mathcal{S}_i$. Thus, we introduce a filtering condition as follows:

$$\frac{C_{\mathcal{S}\backslash\mathcal{S}_i}(\mathbf{w}_m)}{C_{\mathcal{S}_i}(\mathbf{w}_m)} \leq \rho, \;\; C_{\mathcal{S}_i}(\mathbf{w}_m) > 0, \qquad (4)$$

where $C_{\mathcal{S}_i}(\mathbf{w}_m)$ represents the count of the m-gram $\mathbf{w}_m = x_p, \ldots, x_{p+m-1}$ in $\mathcal{S}_i$. $C_{\mathcal{S}\backslash\mathcal{S}_i}(\mathbf{w}_m)$ represents the count of this m-gram $\mathbf{w}_m$ in all source domains except for $\mathcal{S}_i$, and $\rho$ is an m-gram frequency ratio hyperparameter. By applying this criterion, we ensure that $\mathbf{w}_m$ is considered part of the TRF set of $\mathcal{S}_i$ only if its frequency in $\mathcal{S}_i$ is significantly higher than its frequency in other entity types ($\mathcal{S}\backslash\mathcal{S}_i$). Since the number of examples in $\mathcal{S}_i$ is much smaller than the number of examples in $\mathcal{S}\backslash\mathcal{S}_i$, we choose $\rho \geq 1$ but avoid setting it to a large value. This allows for the inclusion of features that are associated with $\mathcal{S}_i$ while also being related to other entity types in the TRF set of $\mathcal{S}_i$. In our experiments, we set $\rho = 3$ and only consider 1-gram texts for simplicity.

Note that the type-related feature extraction module we designed is highly efficient with a computational complexity of $\mathrm{O}(|\mathcal{D}_{in}| \cdot l_{avg} \cdot |\mathcal{T}|)$, where $|\mathcal{D}_{in}|$, $l_{avg}$, and $\mathcal{T}$ represent the number of sentences in the training dataset, the average sentence length, and the entity type set, respectively. This module is able to compute the mutual information criteria in Eq. 4 for all entity types in $\mathcal{T}$ and each token by traversing the tokens in every training sentence just once.

### 4.2 Prompt generation and incorporation

To connect unseen examples with the knowledge within the source domain, we generate and incorporate a unique prompt for each input instance. This process involves a two-stage mechanism: first, relevant TRFs are selected to form prompts, and then these prompts are input into the PLM-based basic model for entity label inference.

**Automatic type-related feature selection.** Given an input sentence $\mathbf{x}$ and the extracted TRF set $\mathcal{R}$,

we formulate the selection of relevant TRFs as a cloze-style task for our PLM-based basic model $\mathcal{M}_b$ (refer to Sec. 3.2). Specifically, we define the following prompt template function $f(\cdot)$ with $K$ [MASK] tokens:

$$f(\mathbf{x}) = \text{``}\mathbf{x}\text{[SEP]type-related features:[MASK]}\ldots\text{[MASK]''.} \qquad (5)$$

By inputting $f(\mathbf{x})$ into $\mathcal{M}_b$, we compute the hidden vector $\mathbf{h}_{\text{[MASK]}}$ of [MASK]. Given a token $r \in \mathcal{R}$, we compute the probability that token $r$ can fill the masked position:

$$p(\text{[MASK]} = r | f(\mathbf{x}))) = \frac{\exp(\mathbf{r} \cdot \mathbf{h}_{\text{[MASK]}})}{\sum_{\tilde{r} \in \mathcal{R}} \exp(\tilde{\mathbf{r}} \cdot \mathbf{h}_{\text{[MASK]}})}, \quad (6)$$

where $\mathbf{r}$ is the embedding of the token $r$ in the PLM $\mathcal{M}_b$. For each [MASK], we select the token with the highest probability as the relevant TRF for $\mathbf{x}$, while discarding any repeating TRFs. For example, as illustrated in Fig. 2, for the sentence "Bolton's spokesperson told CBS News.", the most relevant TRFs include "Spokesmen", "News" and "Corp".

To train $\mathcal{M}_b$ for TRF selection, we define the loss function $\mathcal{L}_{gen}$ as follows:

$$\mathcal{L}_{gen} = \\ -\frac{1}{|\mathcal{D}_{in}|} \sum_{x \in \mathcal{D}_{in}} \sum_{i=1}^{K} \log p(\text{[MASK]}_i = \phi(x, i) | f(\mathbf{x})), \quad (7)$$

where $\phi(x, i)$ denotes the label for the $i$-th [MASK] token in $\mathbf{x}$. To obtain $\phi(x)$, we compute the Euclidean distance between the PLM-based embeddings of each $r \in \mathcal{R}$ and each token in $\mathbf{x}$, selecting the top-$K$ features. Note that our designed automatic selection process effectively filters out irrelevant TRFs for the given input sentence, substantially reducing human interventions in TRF extraction (refer to Sec. 7).

**Prompt incorporation.** To incorporate the entity type information into prompts, we generate a unique prompt given the selected relevant TRFs $\mathcal{R}'(\mathbf{x}) \subseteq \mathcal{R}$ for input $\mathbf{x}$. This is achieved using the following prompt template function $f'(\mathbf{x})$:

$$f'(\mathbf{x}) = \\ \text{``}\mathbf{x}\text{[SEP]}t_1\text{:}\mathcal{R}'(\mathbf{x}, t_1)\text{[SEP]}\ldots\text{[SEP]}t_{|\mathcal{T}|}\text{:}\mathcal{R}'(\mathbf{x}, t_{|\mathcal{T}|})\text{'',}} \quad (8)$$

where $t_i \in \mathcal{T}$ is the entity type name (e.g., PER or ORG). Given sentence $\mathbf{x}$, $\mathcal{R}'(\mathbf{x}, t_i) \subseteq \mathcal{R}'(\mathbf{x})$ represents selected TRFs related to entity type $t_i$. Note that, if $\mathcal{R}'(\mathbf{x}, t_i) = \emptyset$, the entity type name, and relevant TRFs $\mathcal{R}'(\mathbf{x}, t_i)$ are excluded from $f'(\mathbf{x})$. For example, as depicted in

| | # Instances | | | |
|---|---|---|---|---|
| **Dataset** | Train | Dev | Test | **Entity types** |
| CoNLL2003 | 14,987 | 3,466 | 3,684 | 4 |
| OntoNotes | 59,924 | 8,528 | 8,262 | 18 |
| TechNews | - | - | 2,000 | 4 |
| AI | - | - | 431 | 14 |
| Literature | - | - | 416 | 12 |
| Music | - | - | 456 | 13 |
| Politics | - | - | 651 | 9 |
| Science | - | - | 543 | 17 |

Table 1: Statistics of the datasets used.

Fig. 2, the unique prompt $f'(\mathbf{x})$ corresponding to $\mathbf{x}$ = "Bolton's spokesperson told CBS News." can be represented as follows:

$$f'(\mathbf{x}) = \text{"Bolton's spokesperson told CBS News.} \\ \texttt{[SEP]}\text{PER:Spokesmen}\texttt{[SEP]}\text{ORG:News, Corp"}. \quad (9)$$

Then, we input $f'(\mathbf{x})$ into $\mathcal{M}_b$ to recognize entities in the given sentence $\mathbf{x}$.

### 4.3 Joint training

To enable parameter sharing between prompt generation and incorporation, we train our model using a multi-task framework. The overall loss function is defined as follows:

$$\mathcal{L} = \alpha \cdot \mathcal{L}'_{NER} + (1 - \alpha) \cdot \mathcal{L}_{gen}, \quad (10)$$

where $\mathcal{L}'_{NER}$ denotes the normalized loss function for the NER task loss $\mathcal{L}_{NER}$ (refer to Sec. 3.2). $\alpha$ is the weight assigned to $\mathcal{L}'_{NER}$ with prompts as inputs. The weight $1 - \alpha$ is assigned to the loss function $\mathcal{L}_{gen}$ for type-related feature selection. In our experiments, we optimize the overall loss function using AdamW (Loshchilov and Hutter, 2019). Sec. A.1 gives the detailed training algorithm of PLTR.

## 5 Experiments

We aim to answer the following research questions: (RQ1) Does PLTR outperform state-of-the-art fine–tuning methods on the few-shot cross-domain NER task? (Sec. 6.1) (RQ2) Can PLTR be applied to prompt-tuning NER methods? (Sec. 6.2) Micro F1 is adopted as the evaluation metric for all settings.

### 5.1 Datasets

Detailed statistics of both in-domain and out-of-domain datasets are shown in Table 1.

**In-domain dataset.** We conduct in-domain experiments on the CoNLL2003 dataset (Sang and

Meulder, 2003). It consists of text in a style similar to Reuters News and encompasses entity types such as person, location, and organization. Additionally, to examine whether PLTR is extensible to different source domains and entity types, we evaluate PLTR using training data from OntoNotes (Weischedel et al., 2013) (refer to Sec. A.3). OntoNotes is an English dataset consisting of text from a wide range of domains and 18 types of named entities, such as Person, Event, and Date.

**Out-of-domain datasets.** We utilize the OOD dataset collected by Liu et al. (2021), which includes new domains such as AI, Literature, Music, Politics, and Science. The vocabulary overlaps between these domains are generally small, indicating the diversity of the out-of-domain datasets (Liu et al., 2021). Since the model trained on the source domain dataset (CoNLL2003) can only predict person, location, organization, and miscellaneous entities, we assign the label $O$ to all unseen labels in the OOD datasets.

### 5.2 Experimental settings and baselines

We compare PLTR with recent baselines in the following two experimental settings:

**Fine-tuning.** Following Yang et al. (2022), we employ the standard fine-tuning method (Ori) based on two pre-trained models with different parameter sizes: BERT-base, BERT-large, RoBERT-base, and RoBERT-large. All backbone models are implemented using the transformer package provided by Huggingface.[2] For fine-tuning the NER models in a few-shot setting, we randomly select 100 instances per label from the original dataset (CoNLL2003) to ensure model convergence. The reported performance of the models is an average across five training runs.

**Prompt-tuning.** Similar to Yang et al. (2022), we adopt the EntLM model proposed by Ma et al. (2022) as the benchmark for prompt-tuning. The EntLM model is built on the BERT-base or BERT-large architectures. We conduct prompt-based experiments using a 5-shot training strategy (Ma et al., 2022). Additionally, we select two representative datasets, TechNews and Science, for the OOD test based on the highest and lowest word overlap with the original training domain, respectively.

Additionally, we include a recent data augmentation method CF (Zeng et al., 2020) and the state-

---

[2] https://huggingface.co/models

| | In-domain Fine-tuning Results | | | |
|---|---|---|---|---|
| **Backbone** | **Ori** | **CF** | **FactMix** | **PLTR** |
| BERT-base-cased | 54.03 | 77.71 | 80.10 | **82.05\*** |
| BERT-large-cased | 65.38 | 81.11 | 83.04 | **83.75\*** |
| RoBERTa-base | 48.53 | 82.74 | 85.33 | **86.40\*** |
| RoBERTa-large | 65.70 | 85.20 | 86.91 | **88.03\*** |

Table 2: In-domain fine-tuning results (Micro F1) on CoNLL2003. ∗ indicates a statistically significant difference (t-test, p<0.05) when compared to FactMix

of-the-art cross-domain few-shot NER framework FactMix (Yang et al., 2022) as baselines in both of the above settings. Note that, we report the results of FactMix's highest-performing variant for all settings and datasets.

### 5.3 Implementation details

Following Yang et al. (2022), we train all models for 10 epochs and employ an early stopping criterion based on the performance on the development dataset. The AdamW optimizer (Loshchilov and Hutter, 2019) is used to optimize the loss functions. We use a batch size of 4, a warmup ratio of 0.1, and a learning rate of 2e-5. The maximum input and output lengths of all models are set to 256. For PLTR, we search for the optimal loss weight $\alpha$ from {0.1, 0.25, 0.5, 0.75, 0.9}. The frequency ratio hyperparameter $\rho$ is set to 3 for all domains.

## 6 Experimental results

To answer RQ1 and RQ2, we assess the performance of PLTR on both in-domain and cross-domain few-shot NER tasks. This evaluation is conducted in two settings: a fine-tuning setting with 100 training instances per type, and a prompt-tuning setting with 5 training instances per type.

### 6.1 Results on few-shot fine-tuning (RQ1)

Table 2 and 3 show the in-domain and cross-domain performance in the fine-tuning setting, respectively. Based on the results, we have the following observations: (i) PLTR achieves the highest Micro F1 scores for all datasets and settings, indicating its superior performance. For instance, when using RoBERTa-large as the backbone, PLTR achieves an 88.03% and 75.14% F1 score on the CoNLL2003 and TechNews datasets, respectively. (ii) PLTR significantly outperforms the previous state-of-the-art baselines in both in-domain and cross-domain NER. For example, PLTR exhibits a 1.46% and 10.64% improvement over FactMix, on average,

on in-domain and cross-domain datasets, respectively. (iii) Few-shot cross-domain NER is notably more challenging than the in-domain setting, as all methods obtain considerably lower F1 scores. The performance decay in TechNews is smaller than in other domains, due to its higher overlap with the training set. In summary, PLTR demonstrates its effectiveness in recognizing named entities from both in-domain and OOD examples. The use of type-related features (TRFs), along with the incorporation of prompts based on TRFs, are beneficial for in-domain and cross-domain few-shot NER.

### 6.2 Results on few-shot prompt-tuning (RQ2)

To explore the generalizability of PLTR, we report in-domain and OOD results for the prompt-tuning setting in Table 4 and 5, respectively. We obtain the following insights: (i) Due to data sparsity, the overall performance for the prompt-tuning setting is considerably lower than the results of 100-shot fine–tuning. (ii) Even with only 5-shot training instances per entity type, PLTR achieves the highest performance and outperforms the state-of-the-art baselines by a significant margin, demonstrating the effectiveness and generalizability of PLTR. For example, in the in-domain and cross-domain datasets, PLTR achieves an average improvement of 11.58% and 18.24% over FactMix, respectively. In summary, the PLTR framework not only effectively generalizes fine-tuning-based NER methods to unseen domains, but also attains the highest F1 scores in the prompt-tuning setting.

## 7 Analysis

Now that we have answered our research questions, we take a closer look at PLTR to analyze its performance. We examine whether the prompts are designed appropriately. Besides, we study how the number of training samples and selected type-related features influence the performance (Sec. A.2), how PLTR affects representation similarities between the source and target domains, and whether PLTR is extensible to different source domains and entity types (Sec. A.3). Furthermore, we provide insights into the possible factors that limit further improvements.

**Ablation studies.** To investigate the appropriateness of our prompt design, we conduct ablation studies on few-shot cross-domain NER in both fine-tuning and prompt-tuning settings. The results are presented in Table 6. In the "NP" variant, prompts

| | | OOD Fine-tuning Results | | | | | OOD Fine-tuning Results | | | |
|---|---|---|---|---|---|---|---|---|---|---|
| **Backbone** | **Dataset** | **Ori** | **CF** | **FactMix** | **PLTR** | **Dataset** | **Ori** | **CF** | **FactMix** | **PLTR** |
| BERT-base-cased | TechNews | 41.46 | 61.20 | 65.20 | **67.39\*** | Music | 10.46 | 19.33 | 19.49 | **23.86\*** |
| BERT-large-cased | | 52.63 | 67.51 | 69.98 | **70.51\*** | | 12.00 | 19.64 | 19.97 | **27.84\*** |
| RoBERTa-base | | 44.88 | 71.83 | 73.62 | **75.06\*** | | 11.78 | 22.24 | 23.75 | **30.52\*** |
| RoBERTa-large | | 51.76 | 73.11 | 74.89 | **75.14\*** | | 14.44 | 21.13 | 22.93 | **30.26\*** |
| BERT-base-cased | AI | 15.88 | 22.49 | 24.67 | **28.41\*** | Politics | 21.38 | 41.84 | 43.60 | **44.97\*** |
| BERT-large-cased | | 18.62 | 26.00 | 26.25 | **30.25\*** | | 29.77 | 43.37 | 43.84 | **45.85\*** |
| RoBERTa-base | | 18.63 | 32.03 | 32.09 | **33.87\*** | | 26.81 | 44.12 | 44.66 | **47.56\*** |
| RoBERTa-large | | 23.27 | 28.76 | 30.06 | **31.97\*** | | 28.56 | 45.87 | 45.05 | **48.35\*** |
| BERT-base-cased | Literature | 12.85 | 22.89 | 25.70 | **27.39\*** | Science | 12.41 | 25.67 | 29.72 | **31.78\*** |
| BERT-large-cased | | 17.53 | 24.96 | 26.25 | **27.83\*** | | 16.05 | 28.75 | 27.88 | **31.19\*** |
| RoBERTa-base | | 15.05 | 28.21 | 28.89 | **30.80\*** | | 14.17 | 33.33 | 34.13 | **34.87\*** |
| RoBERTa-large | | 19.20 | 25.43 | 26.76 | **31.02\*** | | 17.25 | 31.36 | 32.39 | **35.08\*** |

Table 3: OOD fine-tuning results (Micro F1) over six datasets. ∗ indicates a statistically significant difference (t-test, p<0.05) when compared to FactMix.

| | In-domain Prompt-tuning Results | | | |
|---|---|---|---|---|
| **Backbone** | **EntLM** | **CF** | **FactMix** | **PLTR** |
| BERT-base-cased | 54.00 | 55.61 | 59.19 | **63.50\*** |
| BERT-large-cased | 60.37 | 56.49 | 60.80 | **70.46\*** |

Table 4: In-domain prompt-tuning results (Micro F1) on CoNLL2003. ∗ indicates a statistically significant difference (t-test, p<0.05) when compared to FactMix.

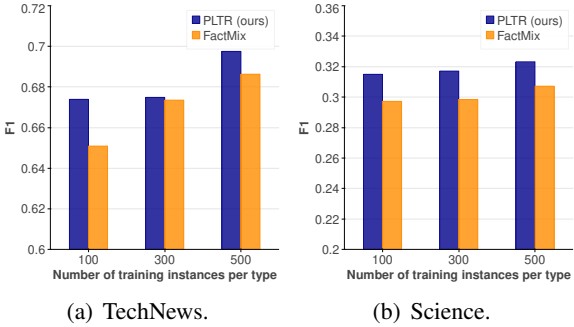

(a) TechNews.  (b) Science.

Figure 3: Influence of training instances on TechNews and Science (BERT-base).

are removed during test-time inference. In this case, the F1 scores across all datasets and settings suffer a significant drop compared to our proposed PLTR. This demonstrates the crucial role of incorporating prompts during both the training and inference processes. In the "RDW" and "REW" variants, prompts are constructed using randomly selected words from the source domain and the given example, respectively. The performance of both the "RDW" and "REW" model variants consistently falls short of PLTR, indicating that PLTR effectively identifies important knowledge from the source domain and establishes connections between unseen examples and the knowledge within the source domain.

Additionally, to explore the efficacy of type-related feature selection (refer to Sec. 4.2), we conducted an evaluation of PLTR (BERT-base) using various frequency ratios $\rho$ (in Eq. 4). The results are presented in Table 7. As the value of $\rho$ increases, TRFs extracted using Eq.4 become less closely associated with the specified entity type but become more prevalent in other types. When the value of $\rho$ is raised from 3 to 9, we observed only a slight decrease in the F1 scores of PLTR. When the value of $\rho$ is raised to 20, the F1 score of PLTR drops, but still surpasses the state-of-the-art

baseline FactMix. These results indicate that PLTR effectively identifies relevant TRFs for OOD examples, considerably mitigating human interventions in the feature extraction process.

**The influence of training samples.** To examine the impact of the number of training samples, we compare the performance of PLTR and FactMix on few-shot cross-domain NER using 100, 300, and 500 training samples per entity type. Fig. 3 displays the results based on the BERT-base-cased model. PLTR exhibits the largest improvements over FactMix when the dataset comprises only 100 training instances per entity type, as opposed to the 300 and 500 training instances scenarios. Furthermore, PLTR consistently outperforms the prior state-of-the-art approach, FactMix, across all experimental settings with varying numbers of training examples, demonstrating its superiority.

**Analysis of sentence similarities.** In our analysis of sentence similarities, we investigate the impact of PLTR on the representation similarities between the source and target domains. We compute the

| Backbone | Dataset | OOD Prompt-tuning Results | | | | Dataset | OOD Prompt-tuning Results | | | |
|---|---|---|---|---|---|---|---|---|---|---|
| | | EntLM | CF | FactMix | PLTR | | EntLM | CF | FactMix | PLTR |
| BERT-base-cased | TechNews | 47.16 | 52.36 | 52.44 | **60.99\*** | Science | 15.70 | 18.32 | 18.62 | **20.90\*** |
| BERT-large-cased | | 52.53 | 48.32 | 48.64 | **61.64\*** | | 15.32 | 15.34 | 16.80 | **19.77\*** |

Table 5: OOD prompt-tuning results (Micro F1) on TechNews and Science. ∗ indicates a statistically significant difference (t-test, p<0.05) when compared to FactMix.

| Dataset | OOD Fine-tuning Results | | | | | OOD Prompt-tuning Results | | | | |
|---|---|---|---|---|---|---|---|---|---|---|
| | FactMix | NP | RDW | REW | PLTR | FactMix | NP | RDW | REW | PLTR |
| TechNews | 65.09 | 66.16 | 66.01 | 66.10 | **67.39** | 52.44 | 54.01 | 55.90 | 56.46 | **60.99** |
| Science | 29.72 | 29.84 | 30.02 | 30.06 | **31.50** | 18.62 | 18.78 | 18,72 | 19.19 | **20.90** |

Table 6: Ablation studies on TechNews and Science.

| Model | Dataset | Frequency ratio $\rho$ | | | | |
|---|---|---|---|---|---|---|
| | | 3 | 5 | 7 | 9 | 20 |
| FactMix | AI | 24.67 | 24.67 | 24.67 | 24.67 | 24.67 |
| PLTR | | **28.41** | **26.36** | **26.61** | **26.42** | **25.70** |
| FactMix | Science | 29.72 | 29.72 | 29.72 | 29.72 | 29.72 |
| PLTR | | **31.78** | **30.07** | **30.11** | **30.58** | **29.91** |

Table 7: Influence of frequency ratio ($\rho$) on AI and Science (BERT-base, fine-tuning).

| Dataset | Sentence length | | | |
|---|---|---|---|---|
| | < 25 | 25–35 | > 35 | Avg. |
| In-domain | 80.12 | 81.25 | 84.12 | 82.05 |
| AI | 25.86 | 24.71 | 29.65 | 28.41 |
| Science | 23.86 | 29.62 | 32.87 | 31.78 |

Table 8: Error analysis on sentence lengths in test sets (BERT-base, fine-tuning).

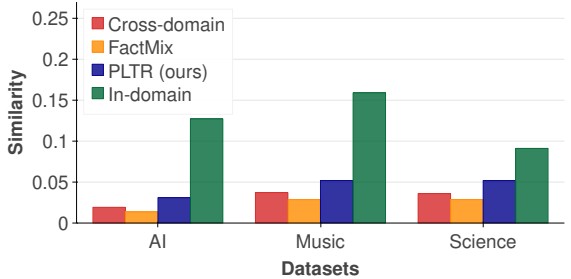

Figure 4: Analysis of sentence similarities on AI, Music, and Science (BERT-base, fine-tuning).

average SBERT similarities for sentence representations in PLTR (BERT-base) between the source and target domains; the results are presented in Fig. 4. With the prompts generated by PLTR, the representation similarities between the source and unseen domains noticeably increase. This is, PLTR facilitates a more aligned and connected representation space, mitigating the gap between the source and target domains.

**Error analysis.** Although our proposed PLTR outperforms state-of-the-art baselines, we would like to analyze the factors restricting further improvements. Specifically, we compare the performance of PLTR (BERT-base) on sentences of different lengths in the test sets of the CoNLL2003 (In-domain), AI, and Science datasets. The results of the standard fine-tuning setting are provided in

Table 8. We observe that the F1 scores of PLTR on sentences with more than 35 words (">35") are substantially higher than the overall F1 scores. In contrast, the F1 scores on sentences with 25 to 35 words ("25–35") or less than 25 words ("<25") consistently fall below the overall F1 scores. This suggests that it may be more challenging for PLTR to select TRFs and generate appropriate prompts with less context.

## 8 Conclusions

In this paper, we establish a new state-of-the-art framework, PLTR, for few-shot cross-domain NER. To capture useful knowledge from the source domain, PLTR employs mutual information criteria to extract type-related features. PLTR automatically selects pertinent features and generates a unique prompt for each unseen example, bridging the gap between domains. Experimental results show that PLTR not only effectively generalizes standard fine-tuning methods to unseen domains, but also demonstrates promising performance when incorporated with prompt-tuning-based approaches. Additionally, PLTR substantially narrows the disparity between in-domain examples and OOD instances, enhancing the similarities of their sentence representations.

## Limitations

While PLTR achieves a new state-of-the-art performance, it has several limitations. First, the number of type-related features for prompt construction needs to be manually preset. Second, PLTR relies on identifying TRFs, which are tokens strongly associated with entity types. Extracting and incorporating more complex features, such as phrases, represents a promising direction for future research. In the future, we also plan to incorporate PLTR with different kinds of pre-trained language models, such as autoregressive language models.

## Ethics statement

The paper presents a prompt-based method for recognizing named entities in unseen domains with limited labeled in-domain examples. However, the constructed prompts and model-predicted results still have a considerable amount of misinformation. Besides, the reliance on black-box pre-trained language models raises concerns. Hence, caution and further research are required prior to deploying this method in real-world applications.

## Acknowledgement

This work was supported by the National Key R&D Program of China (2020YFB1406704, 2022YFC3303004), the Natural Science Foundation of China (62272274, 61972234, 62072279, 62102234, 62202271), the Natural Science Foundation of Shandong Province (ZR2021QF129, ZR2022QF004), the Key Scientific and Technological Innovation Program of Shandong Province (2019JZZY010129), the Fundamental Research Funds of Shandong University, the China Scholarship Council under grant nr. 202206220085, the Hybrid Intelligence Center, a 10-year program funded by the Dutch Ministry of Education, Culture and Science through the Netherlands Organization for Scientific Research, https://hybrid-intelligence-centre.nl, and project LESSEN with project number NWA.1389.20.183 of the research program NWA ORC 2020/21, which is (partly) financed by the Dutch Research Council (NWO).

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

## A  Appendix

### A.1  Training algorithm of PLTR

Algorithm 1 gives the detailed training algorithm of PLTR. To start, we establish a basic model $\mathcal{M}_b$ based on Pre-trained Language Models (PLM) and initialize its parameters $\Theta$ (lines 1-2). To capture knowledge from the source domain, PLTR identifies type-related features using mutual information criteria (line 3). Next, given an input sentence $\mathbf{x} \in \mathcal{D}_{in}$, PLTR automatically selects relevant TRFs $\mathcal{R}'(\mathbf{x}) \subseteq \mathcal{R}$ by formulating the selection process as a cloze-style task for $\mathcal{M}_b$ (line 7). Furthermore, to incorporate entity type information into prompts, PLTR constructs a unique prompt

---

**Algorithm 1** Training Algorithm for PLTR.

---

**Require:** The source dataset $\mathcal{D}_{in}$; the basic model $\mathcal{M}_b$ with parameters $\Theta$; the frequency ratio $\rho$; the number of selected type-related features $K$; the loss weight $\alpha$; the number of epochs $epoch$.

**Ensure:** The extracted type-related features $\mathcal{R}$ and the trained basic model $\mathcal{M}_b'$;

1: Establish the basic model $\mathcal{M}_b$;
2: Initialize model parameters $\Theta$;
3: Extract type-related features $\mathcal{R}$ for all entity types from the source dataset $\mathcal{D}_{in}$ (Eq. 4);
4: **while** $i \leq epoch$ **do**
5:     **for** Sample a batch $\mathcal{X} \subseteq \mathcal{D}_{in}$ **do**
6:         **for** all sentences $\mathbf{x} \in \mathcal{X}$ **do**
7:             Select relevant TRFs $\mathcal{R}'(\mathbf{x})$ for input $\mathbf{x}$ (Eq. 5 and 6);
8:
9:             Transform $\mathbf{x}$ into the prompt template $f'(\mathbf{x})$ (Eq. 8);
10:
11:             Input $f'(\mathbf{x})$ into $\mathcal{M}_b$ for prediction;
12:         **end for**
13:         Update $\Theta$ by optimizing $\mathcal{L}$ (Eq. 10);
14:     **end for**
15: **end while**

---

$f'(\mathbf{x})$ for each input $\mathbf{x}$, and these prompts are then fed into $\mathcal{M}_b$ for entity recognition (lines 8-9). Finally, we iteratively refine the parameters $\Theta$ by jointly optimizing two loss functions: the NER task loss function $\mathcal{L}'_{NER}$ and the TRF selection loss function $\mathcal{L}_{gen}$ (line 11). Note that, during inference, PLTR generates a unique prompt for each sentence within the unseen target domain using extracted TRFs $\mathcal{R}$. In this way, knowledge from the source domain is explicitly integrated into both the training and inference phases.

### A.2  Influence of the number of selected type-related features

We evaluate PLTR based on BERT-base in fine-tuning setting, with the number of selected relevant type-related features $K$ varying from 10 to 60. The results are shown in Fig. 5. Our observations indicate that as the number of type-related features increases, the performance (F1 score) of PLTR initially improves because the model incorporated with more features is able to encode more useful knowledge from the source domain. But notice that the performance drops when the number of type-related features is too large. In our experiments, we set the number of type-related features to 40 on all

| Setting | Model | Dataset | Source: CoNLL2003 | | | | | Source: OntoNotes | | | | |
|---|---|---|---|---|---|---|---|---|---|---|---|---|
| | | | PER | LOC | ORG | MISC | Avg. | PER | LOC | ORG | EVENT | Avg. |
| OOD Fine-tuning Results | FactMix | TechNews | 85.65 | 59.45 | 59.31 | 24.66 | 65.20 | 56.96 | 16.06 | 41.84 | – | 44.11 |
| | PLTR | | **86.00** | **71.34** | **59.93** | **26.25** | **67.39** | **86.14** | **18.33** | **57.71** | – | **65.31** |
| | FactMix | Science | 35.43 | 31.28 | 24.46 | 23.58 | 29.72 | 13.60 | 15.84 | 22.66 | 3.77 | 17.15 |
| | PLTR | | **36.27** | **39.38** | **36.57** | **29.99** | **31.78** | **38.51** | **16.79** | **23.20** | **7.61** | **27.46** |
| OOD Prompt-tuning Results | FactMix | TechNews | 82.88 | 55.05 | 39.82 | 16.12 | 52.44 | 77.66 | 16.20 | 38.92 | – | 53.19 |
| | PLTR | | **87.91** | **56.73** | **42.69** | **29.21** | **60.99** | **79.27** | **20.26** | **42.67** | – | **54.91** |
| | FactMix | Science | 29.87 | 23.17 | 8.47 | 14.30 | 18.62 | 33.14 | 3.96 | 7.54 | 2.90 | 19.83 |
| | PLTR | | **38.51** | **23.19** | **10.46** | **14.33** | **20.90** | **34.98** | **11.83** | **17.27** | **6.67** | **22.32** |

Table 9: Influence of source domains (BERT-base). In TechNews, there are no annotations for "EVENT" entities.

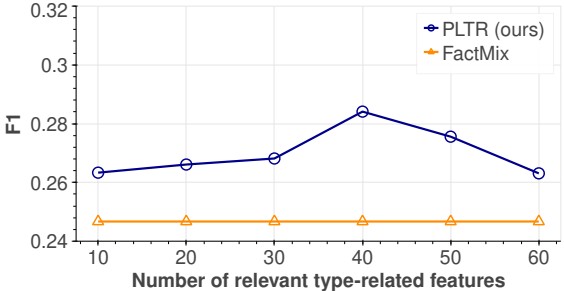

Figure 5: Influence of the number of selected relevant type-related features ($K$) on AI (BERT-base, fine-tuning).

datasets.

### A.3 Influence of source domains

We explore the performance of our proposed PLTR when trained on data from different source domains, i.e., CoNLL2003 and OntoNotes. Results in both the fine-tuning and prompt-tuning settings are shown in Table 9. Our observations indicate that our proposed PLTR consistently outperforms FactMix when trained on different domains. For instance, PLTR achieves an average improvement of 5.42% and 4.22% over FactMix for "LOC" entities when using CoNLL2003 and OntoNotes as source datasets, respectively. This highlights PLTR's capacity to extend to various source domains and entity types.