# OpenReview forum: "Generalizing Few-Shot Named Entity Recognizers to Unseen Domains with Type-Related Features"
_EMNLP/2023/Conference — EMNLP 2023 Findings_

### Official Review · Reviewer_vj1P · 2023-08-03

**Soundness:** 3

**Excitement:**

2: Mediocre: This paper makes marginal contributions (vs non-contemporaneous work), so I would rather not see it in the conference.

**Missing References:**

[1] Wang et al. Gpt-ner: Named entity recognition via large language models.

[2] Li et al. Prompt-based Text Entailment for Low-Resource Named Entity Recognition

**Paper Topic And Main Contributions:**

The paper specifically addresses the problem of few-shot named entity recognition (NER) in low-resource scenarios, where there is limited labeled data available for training models. The paper proposes a novel framework, PLTR, that leverages type-related features (TRFs) to generate unique prompts for each unseen example, thereby reducing the gap between the source and unseen domains. The paper's main contributions are the use of mutual information criteria to extract TRFs, the development of a two-stage framework for few-shot cross-domain NER, and the demonstration of promising performance on several benchmark datasets.

**Questions For The Authors:**

1. Can you provide more details on the limitations or failure cases of the proposed framework? How does the framework perform on different types of data or scenarios?
2. How does the proposed framework compare with other state-of-the-art few-shot cross-domain NER methods? Can you provide a more comprehensive evaluation of the framework's performance against other methods, such as GPT-4?
3. Can you provide more information on the impact of different hyperparameters or design choices on the proposed framework's performance? How can readers best apply the framework to different scenarios?

**Reasons To Accept:**

1. The introduction of a novel approach to few-shot cross-domain NER that leverages type-related features (TRFs) and unique prompts, which could inspire further research in this area.
2. The demonstration of the effectiveness of the proposed framework on several benchmark datasets, which could serve as a baseline for future research in few-shot NER.

**Reasons To Reject:**

1. One potential weakness of this paper is that it does not compare its proposed framework with other state-of-the-art few-shot cross-domain NER methods. A more comprehensive evaluation of the proposed framework's performance against other methods would provide a better understanding of its strengths and limitations.
2. Another potential weakness is that the paper does not provide a detailed analysis of the impact of different hyperparameters or design choices on the proposed framework's performance. A more thorough analysis of these factors would help readers understand how to best apply the framework to different scenarios.
3. The paper does not provide a detailed analysis of the computational complexity of the proposed method, which could be a concern for large-scale applications.
4. The paper does not compare the proposed method to more recent LLM-based NER methods.

**Reproducibility:**

3: Could reproduce the results with some difficulty. The settings of parameters are underspecified or subjectively determined; the training/evaluation data are not widely available.

**Reviewer Confidence:**

4: Quite sure. I tried to check the important points carefully. It's unlikely, though conceivable, that I missed something that should affect my ratings.

---

> ### Author Rebuttal · Authors · 2023-08-28
>
> **Comment 1**: One potential weakness of this paper is that it does not compare its proposed framework with other state-of-the-art few-shot cross-domain NER methods. A more comprehensive evaluation of the proposed framework's performance against other methods would provide a better understanding of its strengths and limitations.
>
> **Response**: Thanks for your comments. Our paper centers around the task of few-shot cross-domain NER, which is different from both cross-domain NER and few-shot NER tasks. In our few-shot cross-domain NER task, no prior knowledge of the target domain is available during training (please refer to Section 3.1 for details). The missing references you provide do not focus on our task but either on few-shot NER or on cross-domain NER. As mentioned in [1] and demonstrated in Section 6, it is not trivial to adapt current prompt-tuning or standard fine-tuning NER methods to effectively address the few-shot cross-domain NER task. FactMix [1] and our PLTR outperform the standard fine-tuning method and EntLM [2] by a large margin.
>
> FactMix [1] is the first work to formulate the few-shot cross-domain NER problem and the latest baseline for this task. Our PLTR significantly outperforms FactMix across a broad range of real-world datasets spanning 6 diverse domains (please refer to Section 6 for details). This demonstrates the strong effectiveness of our proposed framework.
>
> [1] Yang et al. Factmix: Using a few labeled in-domain examples to generalize to cross-domain named entity recognition. In COLING 2022.
> [2] Ma et al. Template-free Prompt Tuning for Few-shot NER. In NAACL 2022.
>
> **Comment 2**: Another potential weakness is that the paper does not provide a detailed analysis of the impact of different hyperparameters or design choices on the proposed framework's performance. A more thorough analysis of these factors would help readers understand how to best apply the framework to different scenarios.
>
> **Response**: Thanks for your comments. Concerning design choices, we evaluate the PLTR framework based on two kinds of pre-trained models with different parameter sizes. PLTR yields the highest F1 scores for all datasets and different design choices concerning backbones (see Section 6). In the fine-tuning setting, PLTR based on RoBERTa-large surpasses other variants in most cases, while in the prompt-tuning setting, PLTR based on BERT-large achieves the best performance in the majority of cases (again, see Section 6).
>
> Regarding the hyperparameters, we analyze the influence of the number of type-related features. The proposed PLTR framework consistently achieves state-of-the-art performance for few-shot cross-domain named entity recognition with different numbers of type-related features (see Figure 5).
>
> Other important hyperparameters whose influence we did not include in the paper due to space constraints are the threshold $\rho$ in Eq. 3 and the loss weight $\alpha$ in Eq. 8. As the results in the following Table 1 show, PLTR automatically selects relevant type-related features from the extracted feature set and is highly robust to the threshold $\rho$. As the results in Table 2 below reveal, the optimal loss weight $\alpha$ in our experiments is 0.5. We will add these additional analyses to the camera-ready version of our paper.
>
> |       | $\rho = 3$ | $\rho = 5$ | $\rho = 7$ | $\rho = 9$ | $\rho = 20$ |
> | ----------- | ----------- | ----------- | ----------- | ----------- | ----------- |
> | **FactMix** | 29.72 | 29.72 | 29.72 | 29.72 | 29.72 |
> | **PLTR** | **31.78** | **30.07** | **30.11** | **30.58** | **29.91** |
> Table 1: Influence of $\rho$ on Science dataset (BERT-base, fine-tuning).
>
> |   | $\alpha = 0.1$ | $\alpha = 0.25$ | $\alpha = 0.5$ | $\alpha = 0.75$ | $\alpha = 0.9$ |
> | - | ----------- | ----------- | ----------- | ----------- | ----------- |
> | **AI** | 24.46 | 24.79 | **28.41** | 25.27 | 24.73 |
> | **Science** | 29.63 | 29.60 | **31.78** | 29.96 | 27.91 |
> Table 2: Influence of $\alpha$ on AI and Science datasets (BERT-base, fine-tuning).
>
>
> **Comment 3**: The paper does not provide a detailed analysis of the computational complexity of the proposed method, which could be a concern for large-scale applications.
>
> **Response**:  Thanks for your comments. The type-related feature extraction module we designed is highly efficient, with a computational complexity of $\rm{O}(N_{train} \cdot l_{avg} \cdot |\mathcal{T}|)$, where $N_{train}$, $l_{avg}$, and $\mathcal{T}$ represent the number of sentences in the training dataset, the average sentence length, and the entity type set, respectively. This module can efficiently compute mutual information criteria in Eq. 3 for all entity types in $\mathcal{T}$ and for each token by traversing the tokens in every training sentence just once. Additionally, our designed prompts can be incorporated with pretrained language models in a plug-and-play manner. Therefore, our proposed PLTR is well-suited for large-scale applications. We will include this analysis of computational complexity in our camera-ready version.
>
> **Comment 4**: The paper does not compare the proposed method to more recent LLM-based NER methods.
>
> **Response**: Thanks for your comment. As mentioned in our response to Comment 1, we have indeed included recent LLM-based baselines in our experiments, such as FactMix and EntLM, which utilize the BERT and RoBERTa architectures.
>
> Recent studies [e.g., 3, 4] indicate that NER models solely based on LLMs, such as the GPT series, tend to significantly underperform when compared to models fine-tuned from smaller pre-trained language models in low-resource scenarios. Therefore, we do not consider these models as baselines in our experiments.
>
> Furthermore, GPT-NER [4] requires both a fine-tuned NER model and GPT-3 to effectively address cross-domain NER and few-shot NER. Consequently, it is unfair to directly compare GPT-NER with other fine-tuning or prompt-tuning approaches.
>
> Besides, GPT-4 is not publicly available and GPT-NER was published in May 2023 (the month before the EMNLP 2023 deadline), making it challenging to reproduce and adapt it to our task settings within the limited time frame.
>
> [3] Gutiérrez et al. Thinking about GPT-3 In-Context Learning for Biomedical IE? Think Again. In EMNLP 2022.
> [4] Wang et al. GPT-NER: Named Entity Recognition via Large Language Models. In Arxiv 2023.
>
> **Question 1**: Can you provide more details on the limitations or failure cases of the proposed framework? How does the framework perform on different types of data or scenarios?
>
> **Response**: Thanks for your interesting questions. We would like to provide an analysis of errors and conduct fine-grained performance comparisons on different entity types, as presented in the following tables.
>
> In terms of error analysis, we examine the framework's performance across different sentence lengths: sentences with less than 25 tokens (<25), sentences with 25 to 35 tokens (25-35), and sentences with more than 35 tokens (>35). The results in the following Table 3 illustrate that the performance of PLTR significantly decreases as sentence length becomes shorter. The reason may be that PLTR faces challenges in selecting relevant type-related features with less context information. Additionally, the results in Tables 4 and 5 demonstrate that PLTR achieves notably lower F1 scores on "MISC" types, likely due to the high diversity of "MISC" entity mentions.
>
> For fine-grained comparisons on different entity types, as the following Tables 4 and 5 show, our proposed PLTR consistently outperforms the SOTA baseline FactMix, highlighting the superiority of our framework.
>
> We appreciate your feedback, and we will incorporate the above-mentioned analyses into our camera-ready version.
>
> |   | <25 | 25 - 35 | >35 | Average
> | -|-|-|-|-|
> | **CoNLL03** | 80.12 | 81.25 | 84.12 | 82.05 |
> | **AI** | 25.86 | 24.71 | 29.65 | 28.41 |
> | **Science** | 23.86 | 29.62 | 32.87 | 31.78 |
> Table 3: Performance on different sentence lengths on CoNLL03, AI, and Science datasets (BERT-base, fine-tuning).
>
> |   | PER | LOC | ORG | MISC | Average |
> |-| ----------- | ----------- | ----------- | ----------- |  ----------- |
> | **FactMix** | 31.38 | 29.57 | 30.04 | 12.28 | 24.67 |
> | **PLTR** | **33.05** | **38.97** | **37.06** | **12.58** | **28.41** |
> Table 4: Fine-grained comparisons on AI datasets (BERT-base, fine-tuning).
>
> |   | PER | LOC | ORG | MISC | Average |
> |-|-|-|-|-|-|
> | **FactMix** | 35.43 | 31.28 | 24.46 | 23.58 | 29.72 |
> | **PLTR** | **36.27** | **39.38** | **36.57** | **29.99** | **31.78** |
> Table 5: Fine-grained comparisons on Science datasets (BERT-base, fine-tuning).
>
> **Question 2**: How does the proposed framework compare with other state-of-the-art few-shot cross-domain NER methods? Can you provide a more comprehensive evaluation of the framework's performance against other methods, such as GPT-4?
>
> **Response**: Thanks for your question. Please see our response to Comments 1 and 4.
>
> **Question 3**: Can you provide more information on the impact of different hyperparameters or design choices on the proposed framework's performance? How can readers best apply the framework to different scenarios?
>
> **Response**: Thanks for your questions. Please see our response to Comment 2.
>
> **About missing references**: Thanks for bringing these references to our attention. Both of the mentioned references focus on cross-domain NER or few-shot NER, that is, on tasks that differ from the problem setting addressed in our paper. Please see our response to Comments 1 and 4 for details.

---

### Official Review · Reviewer_2X7Q · 2023-08-04

**Typos Grammar Style And Presentation Improvements:** N/A
**Soundness:** 3

**Excitement:**

4: Strong: This paper deepens the understanding of some phenomenon or lowers the barriers to an existing research direction.

**Missing References:**

N/A

**Paper Topic And Main Contributions:**

Paper Topic: NER, OOD Generalization.

Main contribution (NLP engineering experiment): The paper proposes a new prompt learning framework for few-shot cross-domain NER to alleviate two limitations in this area: (1) augmentation limited in the source domain; (2) insufficient knowledge transfer. Experiments demonstrate the effectiveness of the proposed approach, where state-of-the-art results are achieved.

**Questions For The Authors:**

N/A

**Reasons To Accept:**

Reasons To Accept:

- Regarding the writing: The paper is well written and easy to follow.
- A mutual information-based approach and an instance-based incorporation approach are designed to improve the approach.
- The authors conducted sufficient results to support their claim.
- The proposed approach achieved statistically promising results in the benchmarks.

**Reasons To Reject:**

Unfortunately, the paper should be **desk rejected** since it violates two submission policies:

- The paper is **over-length** with one figure on Page 9.
- The repository is **non-anonymous** (see the files in __pycache__), where I can see the name in the repo.

**Reproducibility:**

4: Could mostly reproduce the results, but there may be some variation because of sample variance or minor variations in their interpretation of the protocol or method.

**Reviewer Confidence:**

4: Quite sure. I tried to check the important points carefully. It's unlikely, though conceivable, that I missed something that should affect my ratings.

---

> ### Author Rebuttal · Authors · 2023-08-28
>
> **Comment 1**: The paper is over-length with one figure on Page 9.
>
> **Response**: Thanks for pointing this out. The figure on page 9 (i.e., Figure 5) serves to illustrate the impact of the number of type-related features and is included in the appendix section. Our main text spans less than 8 pages, ensuring the paper's overall length is within the specified limits.
>
> **Comment 2**: The repository is non-anonymous (see the files in pycache), where I can see the name in the repo.
>
> **Response**: Thanks for bringing this to our attention. We apologize for any confusion caused. The names appearing in the pycache files are actually server usernames, not the real names of the authors. To be on the safe side, we have removed this information from the repository.

---

### Official Review · Reviewer_NpEC · 2023-08-05

**Soundness:** 3

**Excitement:**

4: Strong: This paper deepens the understanding of some phenomenon or lowers the barriers to an existing research direction.

**Paper Topic And Main Contributions:**

This paper introduces PLTR framework for few-shot cross-domain NER that extracts and leverages type related features based on some mutual information criteria. Basically from a source domain this technique tries to generate some type related features, which they later use in inference example to find best type words. These type words are further used to prompt the model in giving the output NER class. Overall the proposed methodology shows some performance improvement against current baselines.

**Questions For The Authors:**

Do you think it would be extensible to other more challenging datasets and more target categories?

**Reasons To Accept:**

1. A new approach that statistically tries to find the most optimal tokens to transfer knowledge from source to target domain.
2. OOD prompt tuning results show nice improvement over FactMix.

**Reasons To Reject:**

1. Although I really like the idea of finding "type related features" I have some genuine concerns regarding the technique. The statistical approach that was followed may vary quite a bit depending on the target categories. PER, ORG, LOC, and MISC are arguably some of the simplest classes in this category. In more complex technical domains, I am not sure it will be so straightforward to do the type-related feature extraction. In particular, the value of rho = 3, I think it will change wildly depending on some challenging domains, which may require human intervention defeating the purpose. (Addressed in rebuttal)

2. Prompt incorporation with larger number of domains will be even more challenging. With four domains, sure it is simpler. However with datasets like OntoNotes, I guess it will take a lot of tokens to give all the TRFs and classes. This may in turn cause lower performance probably?

3. While I see why the authors mainly using FactMix's evaluation criteria, I think these category of methods should be used in more domains and more categories to properly evaluate the efficacy. Otherwise it is not very easy to objectively say that this method may induce good performance in real life.

**Reproducibility:**

4: Could mostly reproduce the results, but there may be some variation because of sample variance or minor variations in their interpretation of the protocol or method.

**Reviewer Confidence:**

4: Quite sure. I tried to check the important points carefully. It's unlikely, though conceivable, that I missed something that should affect my ratings.

---

> ### Author Rebuttal · Authors · 2023-08-28
>
> **Comment 1**: The statistical approach that was followed may vary quite a bit depending on the target categories. PER, ORG, LOC, and MISC are arguably some of the simplest classes in this category. In more complex technical domains, I am not sure it will be so straightforward to do the type-related feature extraction. In particular, the value of $\rho$ = 3, I think it will change wildly depending on some challenging domains, which may require human intervention defeating the purpose.
>
> **Response**: Thank you for your comment. Our proposed framework isn't confined to specific entity types. For comparability, we followed the approach outlined in FactMix [1] in our experiments, which led us to choose PER, ORG, LOC, and MISC as our target entity types. This choice was made not for simplicity, but because these types are shared between the source and target domains. Our paper does not address unseen entity types in the target domain.
>
> The type-related feature extraction module we designed is highly efficient with a computational complexity of $\rm{O} (N_{train} \cdot l_{avg} \cdot |\mathcal{T}|)$, where $N_{train}$, $l_{avg}$, and $\mathcal{T}$ represent the number of sentences in the training dataset, the average sentence length, and the entity type set, respectively. This module can efficiently compute the mutual information criteria in Eq. 3 for all entity types in $\mathcal{T}$ and for each token by traversing the tokens in every training sentence just once.
>
> Additionally, the value of $\rho$ has limited influence on our proposed PLTR framework. As demonstrated in Section 4.2, PLTR automatically selects relevant type-related features from the extracted feature set while effectively filtering out irrelevant ones. This mechanism renders our framework highly resilient to variations in the value of $\rho$. The following Table 1 shows the influence of $\rho$ on the Science dataset. We can easily find that PLTR is highly robust to the value of $\rho$. We will include this analysis in our camera-ready version.
>
> | | $\rho = 3$ | $\rho = 5$ | $\rho = 7$ | $\rho = 9$ | $\rho = 20$ |
> | ----------- | ----------- | ----------- | ----------- | ----------- | ----------- |
> | **FactMix** | 29.72 | 29.72 | 29.72 | 29.72 | 29.72 |
> | **PLTR** | **31.78** | **30.07** | **30.11** | **30.58** | **29.91** |
> Table 1: Influence of $\rho$ on Science dataset (BERT-base, fine-tuning).
>
> In summary, our proposed type-related feature extraction component is highly efficient, fully automatic, and can be readily extended to accommodate more complex domains with a broader range of entity types.
>
> [1] Yang et al. Factmix: Using a few labeled in-domain examples to generalize to cross-domain named entity recognition. In COLING 2022.
>
> **Comment 2**: Prompt incorporation with larger number of domains will be even more challenging. With four domains, sure it is simpler. However with datasets like OntoNotes, I guess it will take a lot of tokens to give all the TRFs and classes. This may in turn cause lower performance probably?
>
> **Response**: Thanks for your comment. In prompt incorporation, the length of prompts is controllable through the hyperparameter K for Eq. 4 (please see Section 4.2 for details), and it is not determined by either the number of domains or the number of entity types. This parameter allows us to manage prompt length based on the specific requirements of different datasets. To further mitigate the length of prompts, when the selected type-related feature set is null, PLTR removes the corresponding type-related features from the prompts. Therefore, our approach is readily extended to datasets with a larger number of domains and entity types.
>
> Additionally, as Figure 5 shows, our proposed PLTR framework maintains its ability to achieve state-of-the-art performance for few-shot cross-domain named entity recognition with different numbers of type-related features used in prompts.
>
> **Comment 3**: While I see why the authors mainly using FactMix's evaluation criteria,
>  ld be used in more domains and more categories to properly evaluate the efficacy. Otherwise it is not very easy to objectively say that this method may induce good performance in real life.
>
> **Response**: Thank you for your comments. PLTR is evaluated across a diverse range of real-world datasets from six domains, including Politics, Science, Music, Literature, Artificial Intelligence, and Reuter News [2]. This extensive evaluation showcases the strong effectiveness and generalizability of our approach (please see Section 6 for details).
>
> The availability of public datasets for evaluating few-shot cross-domain NER with a substantial number of entity types is currently limited. For example, if we were to use OntoNotes [3] to replace CoNLL03 as the source domain dataset, this would lead to only five shared entity types across source and target domains, namely Event, LOC, ORG, PERSON, and Product. In future work, we intend to construct more complex datasets with a broader range of entity types and address the challenges posed by unseen types in the target domains.
>
> [2] Liu et al. CrossNER: Evaluating Cross-Domain Named Entity Recognition. AAAI 2021.
> [3] Pradhan et al. Towards robust linguistic analysis using OntoNotes. CoNLL 2013.
>
> **Question 1**: Do you think it would be extensible to other more challenging datasets and more target categories?
>
> **Response**: Thanks for the question. PLTR can readily be extended to handle more challenging datasets and a broader range of target categories. This is supported by the following observations:
>
> 1. Our designed type-related feature extraction component is highly efficient and not limited by specific entity types (see our response to Comment 1 for details).
> 2. PLTR showcases the capability to automatically select relevant type-related features, without relying on external data or extensive human interventions  (see our response to Comment 1 for details).
> 3. The prompt length in our PLTR is controllable and PLTR maintains its ability to achieve state-of-the-art performance with different numbers of type-related features (see our response to Comment 2 for details).
> 4. Our experimental results demonstrate the effectiveness of PLTR across real-world datasets from six diverse domains (see our response to Comment 3 for details).

---

### Meta-Review · Area_Chair_ZBu1 · 2023-09-19

**Recommendation:** 2

**Metareview:**

The paper proposes an approach for few-shot cross-domain NER. It mines templates for prompt generation depending on entity types. The type-dependent prompts are shown to improve the effectiveness over several baselines, in particular, FactMix.

The reviewers have found the work interesting. There is not extensive work on few-shot cross-domain NER. This work enriches the literature on this topic.
The reviewers have questioned about the comparison with few-shot NER and cross-domain NER. The authors replied that the comparison is irrelevant. Nevertheless, it would be interesting to compare the proposed method with the existing few-shot methods by considering cross-domain as a single domain. This would show how an in-domain approach can or cannot be generalized to cross-domain scenario.

The proposed prompt mining approach can also be compared to template mining in traditional NER studies based on rules. There is a large body of research work on template mining, which the paper does not refer to. It would also be possible to compare with such traditional approaches in the experiments.

---

### Decision · Program_Chairs · 2023-10-07

**Decision:**

Accept-Findings

**Comment:**

The paper proposes an approach for few-shot cross-domain NER. It mines templates for prompt generation depending on entity types. The type-dependent prompts are shown to improve the effectiveness over several baselines, in particular, FactMix.

The reviewers have found the work interesting. There is not extensive work on few-shot cross-domain NER. This work enriches the literature on this topic.
The reviewers have questioned about the comparison with few-shot NER and cross-domain NER. The authors replied that the comparison is irrelevant. Nevertheless, it would be interesting to compare the proposed method with the existing few-shot methods by considering cross-domain as a single domain. This would show how an in-domain approach can or cannot be generalized to cross-domain scenario.

The proposed prompt mining approach can also be compared to template mining in traditional NER studies based on rules. There is a large body of research work on template mining, which the paper does not refer to. It would also be possible to compare with such traditional approaches in the experiments.